# Changes in Physical and Chemical Parameters of Beetroot and Carrot Juices Obtained by Lactic Fermentation

Emilia Janiszewska-Turak [1,*], Katarzyna Pobiega [2], Katarzyna Rybak [1], Alicja Synowiec [2], Łukasz Woźniak [3], Urszula Trych [4], Małgorzata Gniewosz [2] and Dorota Witrowa-Rajchert [1,*]

[1]  Department of Food Engineering and Process Management, Institute of Food Sciences, Warsaw University of Life Sciences—SGGW, 02-787 Warsaw, Poland; katarzyna_rybak@sggw.edu.pl

[2]  Department of Food Biotechnology and Microbiology, Institute of Food Sciences, Warsaw University of Life Sciences—SGGW, 02-787 Warsaw, Poland; katarzyna_pobiega@sggw.edu.pl (K.P.); alicja_synowiec@sggw.edu.pl (A.S.); malgorzata_gniewosz@sggw.edu.pl (M.G.)

[3]  Department of Food Safety and Chemical Analysis, Institute of Agricultural and Food Biotechnology, 02-532 Warsaw, Poland; lukasz.wozniak@ibprs.pl

[4]  Department of Fruit and Vegetable Product Technology, Institute of Agricultural and Food Biotechnology, 02-532 Warsaw, Poland; urszula.trych@ibprs.pl

*  Correspondence: emilia_janiszewska_turak@sggw.edu.pl (E.J.-T.); dorota_witrowa_rajchert@sggw.edu.pl (D.W.-R.); Tel.: +48-225-937-366 (E.J.-T.); +48-225-937-568 (D.W.-R.)

**Abstract:** Fermented foods have gained popularity recently, especially lactofermented vegetable juice products that provide a rich source of nutrients. This study analyzed the properties of fermented beetroot, carrot and beetroot-carrot juices fermented with bacterial strains *Lactiplantibacillus plantarum* or *Levilactobacillus brevis*. Density, extract, dry matter content, pH, total acidity, pigments (betalain or carotenoids), color and LAB count were measured. The results showed that the LAB strains used were capable of fermenting the analyzed juices. It was proven that *Lactiplantibacillus plantarum* was the better strain for fermentation of vegetable juice. This might have been influenced by the fermentation temperature and the duration time. The highest number of lactic acid bacteria was observed for carrot juice fermented with *L. brevis* on the 4th day (9.41 log CFU/mL), while for *L. plantarum* the highest numbers were observed on the 5th–6th day (8.9–9.1 log CFU/mL). For beetroot and its mixture with carrot juices, higher results were observed on the 5th day for *L. plantarum* (9.0 and 8.3 log CFU/mL, respectively). In most variants of this process, fermentation can be completed after 4–5 days. During the fermentation of carrot and mixed juice slow degradation of carotenoids was observed, while in mixed juice an increase of red pigment (betanin) was observed. In beetroot juices huge degradation was observed for fermentation with *L. brevis*, while the second bacteria kept the same level of red pigment almost until the last day, while yellow decreased from the 3rd day. It can be concluded that the mixed juice was most stable according to the bacteria count and the pigment content. In our opinion, *L. plantarum* was better for the fermentation of juices than *L. brevis*. However, in future tests the time of fermentation can be shortened to 5 days and other LAB strains can be tested.

**Keywords:** beetroot; carrot; juice; lactic acid fermentation; carotenoids; betalain; color

## 1. Introduction

Lactic fermentation involves the decomposition of organic matter, through the action of enzymes, into simple compounds; anaerobic conditions for the process are a prerequisite. The process is conducted by various types of lactic acid bacteria, which use enzymes to convert simple sugars and disaccharides into lactic acid and other compounds [1,2]. Bacteria from the genera *Bifidobacterum*, *Lactobacillus*, *Leuconostoc* and *Lactococcus* (e.g., *Lacticaseibacillus paracasei* subsp. *paracasei*, *Lacticaseibacillus rhamnosus*, *L. plantarum*) are most commonly used. Lactic acid bacteria (LAB) are the key organisms in LA fermentation, while they produce lactic acid, the main product of carbohydrate fermentation [3,4].

Lactic acid bacteria are a group of microorganisms distinguished by their metabolic properties. They obtain energy through the process of lactic fermentation. A distinction is made between homofermentative and heterofermentative bacteria [5]. Lactic acid bacteria and their metabolites are recognized as safe (GRAS) and are used in food as natural preservatives. Thanks to LAB, it is possible to inhibit the growth of pathogenic microorganisms in food production or extend their shelf life [6,7]. Lactic fermentation bacteria is a group that has been distinguished for sugar fermentation under microaerophilic and anaerobic conditions with lactic acid [4,8]. The metabolism of sugars is a common characteristic linking these microorganisms, as well as the fact that they are Gram positive and catalase negative, do not spore and tolerate low pH. Lactic bacteria have either the shape of bacilli (e.g., *Lactobacillus*) or cocci (*Lactococcus*, *Streptococcus*), while *Bifidobacterium* can be Y- or V-shaped [9,10]. The therapeutic properties of bacteria are determined by the production of bacteriocins, the synthesis of hydrogen peroxide, the property of lowering the pH of the gastrointestinal tract, and the production of low molecular weight metabolic products [11]. The most commonly used LAB strain is *Lactiplantibacillus plantarum*, which belongs to the *Lactobacillaceae* family. They are rod-shaped bacteria with cells approximately 3–8 µm long and 0.9–1.2 µm wide. They are classified as thermophiles and can grow at a pH range of 3.4–8.8 [12,13]. *L. plantarum* belongs to the group of facultative heterofermentative bacteria. These species are capable of synthesizing bioactive compounds such as exopolysaccharides, bacteriocins, γ-aminobutyric acid, folic acid and riboflavin, and thus can be used for food preservation [14]. Another valuable strain is the *Levilactobacillus brevis*, which belongs to the *Lactobacillaceae* family. These are Gram positive, rod-shaped bacteria. They carry out heterofermentation during which they synthesize lactic acid, carbon dioxide and ethanol [15]. These bacteria belong to mesophiles. They show optimal growth for an environment where the pH is in the range of 4–6 [16]. *L. brevis* has been classified as a bacterium whose consumption improves human immune function [14].

The bottom of the nutritional pyramid mainly contains vegetables and fruits. Therefore, they are a desirable as substrate for fermentation. Fermented fruit and vegetable juices are valued by consumers despite their strong flavor, for their high nutritional value, the content of polyphenols, flavonoids, antioxidants, and minerals, as well as low sodium, cholesterol and fat. Fermented vegetable/fruit juices contain high amounts of components such as sugars and short-chain organic acids [17]. The most popular fermented vegetable drinks are carrot and beetroot juices [18].

Beetroot and carrot are among the most commonly harvested vegetables in Poland. For this reason, we selected beetroot and carrot for the study, as well as a mixture of both juices. The selection was also based on the composition of information about each juice and its potential health properties. In addition, the choice was made for juices made from the most available in the country. Beetroot juices have a high fiber content of 0.7–1.1 g fresh weight, folic acid, vitamins A, C, E, K and B, as well as mineral salts such as zinc, iron, sodium potassium, magnesium, phosphorus and calcium. Beetroot juice has a cytotoxic effect, which is the agent's ability to disrupt cell function against cancer cells. In addition, it is believed to have health-promoting effects, such as anti-stress and anti-atherosclerosis, lowering cholesterol levels and regulating blood clotting. As a result of fermentation, the number of lactic acid bacteria increased in beetroot juice [19]. In a study by Klewicka et al. [20], LAB from the genera *Leuconostoc*, *Lactobacillus* and *Pediococcus* were found. The fermentation process involves changes in pH and can also affect the content of betalain pigments, thus indirectly changing the color parameters [21].

On the other hand, carrot juices are appreciated for their content of many biologically active compounds, as well as for their vitamin content such as vitamins A, C and B, β-carotene content, carotenoids and elements such as zinc, calcium, sodium, potassium, iron, magnesium, copper and lead [22,23]. Michalczyk et al. [24] concluded that carrot juices can be used as a substitute for fresh vegetables due to their high stability in terms of nutrient content during storage. Microbiologically Gientka et al. [25] found mesophilic

microorganisms in carrot juices ranging from 3.1 to 10.0 log CFU/mL, while no pathogenic microflora was detected.

The aim of this study was to analyze the physical and chemical properties of fermented beetroot, carrot and beetroot-carrot juices. The juices obtained from beetroot and carrot were fermented with bacterial strains of *L. plantarum* or *L. brevis*. The levels of change in extract, density, pigment content and type, and color of juices were tested. To better understand the fermentation process, a daily analysis of pH, total acidity, and the microbial count was made. The research hypothesis assumes the influence of varying process conditions (pH, acidity) on pigment degradation. Furthermore, it was assumed that selected bacterial strains have identical effects on the mentioned changes.

## 2. Materials and Methods

### 2.1. Materials

Beetroot (*Beta vulgaris*) and carrot (*Daucus carota*) were purchased from the local market (Warsaw, Poland). Two different strains were used as inoculum for fermentation: *Lactiplantibacillus plantarum* ATCC 4080 (LP) and *Levilactobacillus brevis* DSMZ 20053 (LB). The strains were obtained from the American Type Culture Collection (ATCC, Manassas, VA, USA) and German Collection of Microorganisms and Cell Cultures GmbH (DSMZ, Braunschweig, Germany).

### 2.2. Technological Treatment

#### 2.2.1. Juice Preparation

The juice was obtained after pressing raw vegetables in commercial single-screw juicer NS-621CES (Kuvings, Daegu, Republic of Korea). In this device separation of juice from pomace took place. The juice was used in this study. Three types of juice were used for further research: beetroot, carrot and a mixture of both in proportion 1:1 *v/v*. All juices were pasteurized at 80 °C for 30 min in an autoclave PHCBI MLS—3751 (PHC Europe B.V., Etten-Leur, The Netherlands) to remove the autochthonous microflora. The juices were then cooled to room temperature (25 °C ± 2).

#### 2.2.2. Juice Fermentation Process

Before the addition of inoculum, 2% m/v NaCl was added directly to the cooled juice. Then inoculum in the amount of 1% of the juice volume was added. This amount of inoculum matched the $1 \times 10^8$ CFU/mL of the bacterial count. The 50 mL jars were closed and placed in an incubator (BD-S115, Binder, Tuttlingen, Germany) with temperature of 28 °C for 7 days. Analysis of each parameter was made on each day of the fermentation process. For this reason, 7 sterile 50 mL jars were prepared for each juice for one inoculum strain. All experiments were made in triplicate.

### 2.3. Analytical Method

#### 2.3.1. Solid Soluble Content and Density of the Juices

The solid soluble content in juices was measured with a Pocket Refractometer PAL-3 (ATAGO Instruments, Tokyo, Japan). Density was measured using a Densito 30 PX densito-meter (Mettler Toledo, Schwerzenbach, Switzerland). All measurements were performed in triplicate.

#### 2.3.2. pH Measurement

The pH value was analyzed using a SevenCompact s210 pH meter (Mettler Toledo, Schwerzenbach, Switzerland). Three analyses were made for each juice.

#### 2.3.3. Total Acidity

The total acidity of the juices was tested by the potentiometric method. Solution of 0.1 M sodium hydroxide was added to the sample until a pH of 8.1 was reached. The

result was calculated in g of lactic acid per 100 g of juice dry matter. The measurement was performed in triplicate.

For the measurement of total acidity, the titration method with 0.1 M NaOH was used. This measurement was made in triplicate for each sample.

### 2.3.4. Color Parameters

The color analysis of the juices was made in CR-5 (Konica Minolta Sensing Inc., Osaka, Japan) in the CIE L*a*b* system. Parameters used were calibration of black and white color, illuminant D65, angle of observation 2°. All measurements were performed with 5 repetitions.

### 2.3.5. Indication of the Number of Lactic Acid Bacteria

For the enumeration of living cells, the method of the total count by pour plate was used. Juice was diluted with sterile saline (0.85% NaCl, Biomaxima, Poland). The samples were put onto plates with de Man Rogosa and Sharpe Agar (MRS, Biomaxima, Poland) and incubated at $28\,^{\circ}C \pm 1\,^{\circ}C$ for $48\,h \pm 4\,h$. The number of grown colonies was counted (ProtoCOL 3—Automatic colony counting and zone measuring, Synbiosis, Frederick, MD, USA) and recorded as log CFU/mL. The samples were analyzed in triplicate.

### 2.3.6. Bacteria morphology

After 24 h of culture, the bacteria were stained with crystal violet. The image was taken at 600 magnification on an OPTA-TECH microscope (Warsaw, Poland). OptaView7 software was used to measure cell length and width.

### 2.3.7. Pigment Content

The daily content of betalain and carotenoids was measured by spectrophotometric methods.

1.    Betalains

The calculation was made by the spectrophotometric method presented by Janiszewska-Turak et al. [26]. Juice (0.5 g) was diluted to a volume of 50 mL with phosphate buffer pH 6.5. Extractions were carried out using a Multi Reax mechanical stirrer (Heidolph, Schwabach, Germany) for 10 min. The solution was centrifuged ($5000\times g$, 5 min) and the absorbance at 438, 538 and 600 nm was measured. The content of red betanins and yellow vulgaxanthines in 100 g of juice dry substance was calculated. The analysis was performed in triplicate.

2.    Carotenoids analysis

The total carotenoid content (TCC) was measured according to methodology [27,28] based on spectrophotometric measurements. A specific wavelength used for carotenoid detection was used, 450 nm (Spectronic 200; Thermo Fisher Scientific Inc., Waltham, MA, USA). The juice was extracted twice with acetone and petroleum ether. The blank absorbance was measured at 450 nm for the ether. The results were showed as mg β-carotene/100 g of juice dry substance. The analysis was performed in triplicate.

### 2.3.8. Pigment Identification

Liquid chromatography was used to identify the pigments in fresh juice and juice fermented on the 3rd and 7th day.

Betalains were analyzed with the method presented by Janiszewska-Turak et al. [29] and carotenoids were analyzed with the method of Janiszewska-Turak et al. [30].

For the determination of betalains, 1 g of juice was extracted with a mixture of 0.2% formic acid and acetonitrile. Waters SunFire C8 column (5 μm, 250 × 4.6 mm) with a mobile phase flow (0.2% formic acid, acetonitrile) of 1 mL/min in a gradient was used for separation.

Carotenoid analysis was conducted with the extraction by acetone/hexane 1:1 (*v/v*) containing butylated hydroxytoluene (0.5 g/L) added in three portions of 10 mL to the

5 mL of sample. For water residue removal, the organic phases were washed with 50 g/L of sodium chloride solution. Before analysis samples were evaporated in vacuo, as dissolved in isopropanol and membrane filtered (0.2 μm). Measurements were performed in triplicate.

### 2.4. Statistical Treatment

The data in tables and figures are expressed as the mean ± standard deviation. R platform was used for Pearson's rank correlation analysis ($p < 0.05$) as well as for making plots.

Statistical data was evaluated using the Statistica 13 software (TIBCO Software Inc., Palo Alto, CA, USA). To assess the significance of the differences, analysis of variance such as the one-way ANOVA and HDS Tuckey test were carried out at the $p < 0.05$ level of significance.

## 3. Results

### 3.1. Fermentation

During a seven-day fermentation process of vegetable juices, the count of lactic bacteria was measured daily. Figure 1 displays the count of lactic acid bacteria in fermented vegetable juices on the following days of fermentation. The initial count of lactic acid bacteria in fermented juices ranged from 6.18 to 6.78 log CFU/mL (Figure 1). No differences were observed between the juice inoculation variants in fermented beetroot and carrot-beetroot juice. However, carrot juice contained a higher number of *L. brevis* bacteria in the first few days of fermentation, and a higher number of *L. plantarum* was observed on the sixth day of fermentation. In carrot juice inoculated with *L. brevis* on the third day of fermentation, an increase in the number of bacteria from 7.96 to 9.19 log CFU/mL was observed. However, it decreased from 8.67 to 7.34 log CFU/mL on the sixth day. In carrot juice inoculated with *L. plantarum*, the number of bacteria increased from 6.33 to 8.39 log CFU/mL on the fourth day of fermentation. In beetroot juice inoculated with *L. brevis*, the number of bacteria decreased of two log cycles during fermentation. In beetroot juice inoculated with *L. plantarum*, an increase in the number of bacteria was observed on the fifth day of fermentation from 7.39 to 8.76 log CFU/mL. In carrot-beetroot juice inoculated with *L. brevis*, an increase in the number of bacteria occurred over the fermentation time by about two log cycles. The number of bacteria in carrot-beetroot juice inoculated with *L. plantarum* increased on the second day of fermentation from 6.94 to 7.97 log CFU/mL and decreased on the seventh day from 8.99 to 7.84 log CFU/mL. An increase in the number of bacteria in the juice samples was observed between day zero and the third day of fermentation.

On the last day of fermentation, the highest concentration of *L. brevis* bacteria was found in beetroot juice—8.73 log CFU/mL. In conclusion, the highest number of lactic acid bacteria was observed on fermentation's third and fourth days. In most variants of this process, after this time, fermentation is completed. Juices fermented with the use of *L. brevis* were characterized by a greater number of bacteria in the initial days of the process, while in the final days, a greater number of *L. plantarum* bacteria was observed in tested juices.

The results of this study are similar to those described by Garcia et al. [31]. They showed that the number of *L. plantarum* bacteria in pumpkin puree on the fourth day of fermentation was between 7 and 8 log CFU/mL. In the study by Janiszewska-Turak et al. [30], the number of LABs in fermented beetroot juice was determined to range from 6.5 to 8.0 log CFU/mL. Similar results were obtained by Chabłowska et al. [32], where the number of *L. plantarum* was also higher than *L. brevis*. *L. plantarum* count ranged from $3.33 \times 10^9$ to $7.33 \times 10^9$ CFU/mL depending on the species, while *L. brevis* was $1.6 \times 10^9$ CFU/mL Chabłowska et al. [32].

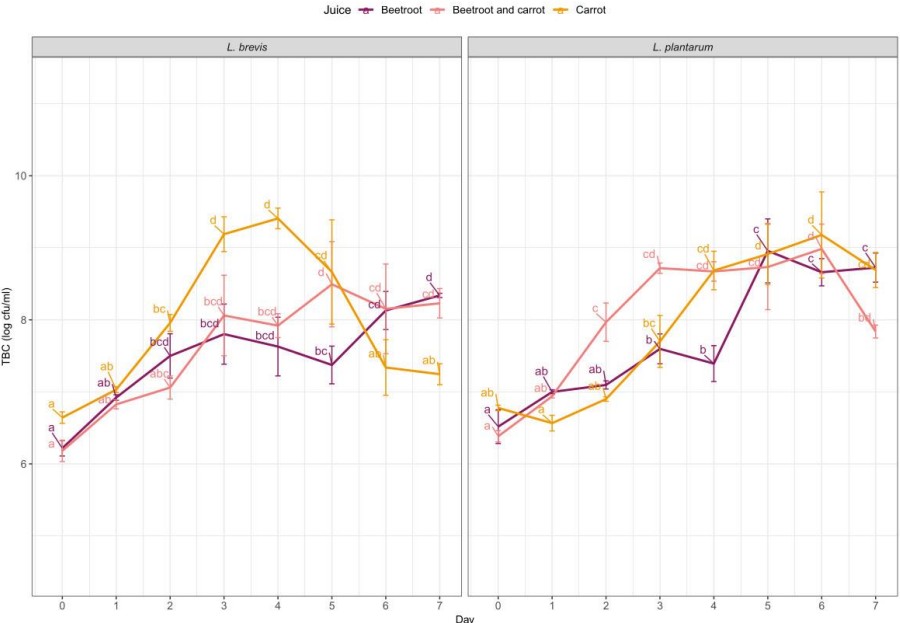

**Figure 1.** The number of lactic acid bacteria during fermentation; different indexes (a, b, c and d) placed at each point, for separate series means statistically significant differences at the level of *p* < 0.05.

### 3.2. Physical Properties of Juices

The highest extract value was recorded for beetroot juice and the lowest for carrot juice (Tables 1 and 2).

**Table 1.** Selected physical properties of juices fermented with *Lactiplantibacillus plantarum*.

| | | | Beetroot Juice | | | | | | | |
|---|---|---|---|---|---|---|---|---|---|---|
| **Day** | | **Fresh** | *Lactiplantibacillus plantarum* | | | | | | | |
| | | **0** | **1** | **2** | **3** | **4** | **5** | **6** | **7** | |
| Extract (°Brix) | | 10.4 ± 0.2 a | 10.9 ± 0.1 b | 10.9 ± 0.1 b | 10.9 ± 0.1 b | 10.8 ± 0.1 b | 11.0 ± 0.1 b | 10.6 ± 0.1 a | 10.7 ± 0.2 a |
| Density (kg/m³) | | 1041 ± 0 a | 1041 ± 1 a | 1040 ± 1 a | 1045 ± 0 cd | 1043 ± 1 b | 1046 ± 0 d | 1045 ± 0 bc | 1044 ± 0 bcd |
| Color coefficients | L* | 2.5 ± 0.1 a | 2.4 ± 0.1 a | 2.8 ± 0.1 b | 2.8 ± 0.1 b | 3.1 ± 0.1 c | 4.1 ± 0.1 d | 3.1 ± 0.1 c | 3.1 ± 0.0 c |
| | a* | 7.8 ± 0.2 a | 7.6 ± 0.4 a | 9.7 ± 0.2 c | 10.6 ± 0.3 d | 10.2 ± 0.2 cd | 12.5 ± 0.2 e | 10.4 ± 0.2 d | 8.6 ± 0.2 b |
| | b* | 1.4 ± 0.1 a | 1.6 ± 0.3 ab | 1.8 ± 0.1 b | 1.9 ± 0.1 b | 1.6 ± 0.2 ab | 2.4 ± 0.1 c | 1.9 ± 0.1 b | 1.7 ± 0.1 b |
| | | | A mix of beetroot and carrot juice | | | | | | | |
| **Day** | | **Fresh** | *Lactiplantibacillus plantarum* | | | | | | | |
| | | **0** | **1** | **2** | **3** | **4** | **5** | **6** | **7** | |
| Extract (°Brix) | | 10.0 ± 0.2 a | 10.5 ± 0.1 bc | 10.6 ± 0.2 bc | 10.6 ± 0.1 bc | 10.5 ± 0.3 bc | 10.3 ± 0.2 b | 10.2 ± 0.1 c | 10.4 ± 0.1 b |
| Density (kg/m³) | | 1033 ± 1 b | 1030 ± 2 a | 1030 ± 0 a | 1036 ± 0 de | 1043 ± 0 f | 1035 ± 0 cd | 1037 ± 0 e | 1034 ± 0 c |
| Color coefficients | L* | 8.1 ± 0.0 ab | 8.8 ± 0.3 c | 8.8 ± 0.1 cd | 8.2 ± 0.1 b | 8.4 ± 0.1 b | 9.1 ± 0.1 d | 7.9 ± 0.1 a | 8.9 ± 0.1 cd |
| | a* | 15.5 ± 0.0 a | 24.6 ± 0.1 f | 24.5 ± 0.2 ef | 24.9 ± 0.2 f | 24.1 ± 0.3 de | 23.0 ± 0.1 c | 23.8 ± 0.1 d | 22.3 ± 0.2 b |
| | b* | 6.3 ± 0.0 a | 7.4 ± 0.0 d | 7.2 ± 0.1 d | 7.0 ± 0.3 d | 6.3 ± 0.1 ba | 7.0 ± 0.1 cd | 6.7 ± 0.1 bc | 7.8 ± 0.2 e |
| | | | Carrot juice | | | | | | | |
| **Day** | | **Fresh** | *Lactiplantibacillus plantarum* | | | | | | | |
| | | **0** | **1** | **2** | **3** | **4** | **5** | **6** | **7** | |
| Extract (°Brix) | | 9.7 ± 0.1 a | 10.5 ± 0.0 b | 10.6 ± 0.1 bc | 10.8 ± 0.0 c | 10.6 ± 0.3 bc | 10.5 ± 0.3 b | 10.4 ± 0.1 b | 10.9 ± 0.1 c |
| Density (kg/m³) | | 1033 ± 1 bc | 1033 ± 1 bc | 1031 ± 2 a | 1031 ± 0 ab | 1030 ± 1 a | 1036 ± 0 d | 1035 ± 1 cd | 1032 ± 1 ab |
| Color coefficients | L* | 33.7 ± 0.0 cd | 34.1 ± 0.5 de | 34.6 ± 0.1 e | 32.5 ± 0.2 a | 32.8 ± 0.2 ab | 35.3 ± 0.1 f | 33.2 ± 0.1 bc | 33.6 ± 0.1 c |
| | a* | 23.0 ± 0.0 a | 23.5 ± 0.1 b | 23.5 ± 0.0 b | 25.3 ± 0.3 c | 25.1 ± 0.3 c | 27.1 ± 0.1 e | 26.0 ± 0.3 d | 25.4 ± 0.2 c |
| | b* | 36.5 ± 0.2 b | 35.5 ± 0.1 a | 35.6 ± 0.0 a | 35.4 ± 0.5 a | 35.2 ± 0.3 a | 38.4 ± 0.1 c | 36.8 ± 0.3 b | 36.6 ± 0.2 b |

The placement of varying indexes such as a, b, c, d, e and f alongside the exact parameter values (as read for each row) indicates statistically significant differences in values at a level of *p* < 0.05. L* is color coefficient brightness, a* is color coefficient redness (+values)/greenness (-values); b* is color coefficient yellowness (+values)/ blueness (-values).

**Table 2.** Selected physical properties of juices fermented with *Lactiplantibacillus brevis*.

| | | | | | **Beetroot Juice** | | | | |
|---|---|---|---|---|---|---|---|---|---|
| **Day** | | **Fresh** | | | *Levilactobacillus brevis* | | | | |
| | | **0** | **1** | **2** | **3** | **4** | **5** | **6** | **7** |
| Extract (°Brix) | | 10.4 ± 0.2 [c] | 10.9 ± 0.3 [a] | 10.9 ± 0.2 [ab] | 10.8 ± 0.2 [ab] | 10.8 ± 0.3 [ab] | 10.6 ± 0.1 [bc] | 10.3 ± 0.1 [cd] | 10.1 ± 0.1 [d] |
| Density (kg/m$^3$) | | 1041 ± 0 [a] | 1040 ± 1 [a] | 1040 ± 3 [a] | 1041 ± 1 [a] | 1043 ± 0 [a] | 1041 ± 1 [a] | 1053 ± 0 [b] | 1041 ± 0 [a] |
| Color coefficients | L* | 2.5 ± 0.1 [a] | 2.5 ± 0.1 [a] | 2.8 ± 0.2 [b] | 3.7 ± 0.1 [c] | 5.3 ± 0.0 [d] | 5.1 ± 0.1 [e] | 4.0 ± 0.0 [f] | 7.5 ± 0.1 [g] |
| | a* | 7.8 ± 0.2 [b] | 7.9 ± 0.3 [a] | 7.4 ± 0.3 [a] | 10.5 ± 0.2 [d] | 12.9 ± 0.1 [e] | 10.1 ± 0.0 [c] | 10.9 ± 0.2 [d] | 9.7 ± 0.0 [c] |
| | b* | 1.4 ± 0.1 [a] | 1.5 ± 0.2 [a] | 1.0 ± 0.1 [a] | 1.8 ± 0.1 [b] | 3.6 ± 0.1 [d] | 3.3 ± 0.1 [d] | 2.5 ± 0.1 [c] | 5.7 ± 0.1 [e] |
| | | | | | **A mix of beetroot and carrot juice** | | | | |
| **Day** | | **Fresh** | | | *Levilactobacillus brevis* | | | | |
| | | **0** | **1** | **2** | **3** | **4** | **5** | **6** | **7** |
| Extract (°Brix) | | 10.0 ± 0.2 [a] | 10.2 ± 0.1 [b] | 10.5 ± 0.1 [a] | 10.7 ± 0.0 [c] | 10.7 ± 0.1 [c] | 10.7 ± 0.2 [c] | 10.6 ± 0.1 [c] | 10.4 ± 0.1 [abc] |
| Density (kg/m$^3$) | | 1033 ± 1 [ab] | 1033 ± 2 [abc] | 1031 ± 0 [a] | 1035 ± 0 [a] | 1034 ± 0 [a] | 1032 ± 0 [a] | 1033 ± 1 [ab] | 1045 ± 0 [d] |
| Color coefficients | L* | 8.1 ± 0.0 [a] | 8.2 ± 0.1 [a] | 8.2 ± 0.1 [a] | 8.6 ± 0.4 [ab] | 8.6 ± 0.5 [ab] | 8.9 ± 0.3 [bc] | 9.3 ± 0.4 [cd] | 9.6 ± 0.1 [d] |
| | a* | 15.5 ± 0.0 [a] | 19.6 ± 0.2 [c] | 19.8 ± 0.2 [c] | 22.4 ± 0.3 [f] | 21.5 ± 0.1 [e] | 18.8 ± 0.1 [b] | 21.1 ± 0.2 [d] | 21.0 ± 0.1 [d] |
| | b* | 6.3 ± 0.0 [a] | 6.0 ± 0.2 [c] | 6.0 ± 0.2 [c] | 8.2 ± 0.5 [f] | 8.0 ± 0.6 [e] | 6.6 ± 0.2 [b] | 8.9 ± 0.3 [d] | 9.0 ± 0.1 [d] |
| | | | | | **Carrot juice** | | | | |
| **Day** | | **Fresh** | | | *Levilactobacillus brevis* | | | | |
| | | **0** | **1** | **2** | **3** | **4** | **5** | **6** | **7** |
| Extract (°Brix) | | 9.7 ± 0.1 [a] | 10.8 ± 0.2 [cd] | 10.7 ± 0.1 [bc] | 11.0 ± 0.1 [d] | 10.8 ± 0.1 [cd] | 10.4 ± 0.1 [bc] | 10.5 ± 0.0 [bc] | 10.3 ± 0.1 [b] |
| Density (kg/m$^3$) | | 1033 ± 1 [e] | 1029 ± 0 [d] | 1022 ± 0 [b] | 1025 ± 1 [c] | 1022 ± 0 [b] | 1028 ± 0 [d] | 1035 ± 1 [f] | 1018 ± 1 [a] |
| Color coefficients | L* | 33.7 ± 0.0 [c] | 37.3 ± 0.1 [g] | 35.7 ± 0.2 [f] | 34.1 ± 0.2 [d] | 31.7 ± 0.2 [d] | 33.2 ± 0.0 [b] | 34.6 ± 0.0 [e] | 32.0 ± 0.2 [a] |
| | a* | 23.0 ± 0.0 [a] | 23.5 ± 0.2 [a] | 25.0 ± 0.3 [b] | 25.9 ± 0.3 [c] | 22.8 ± 0.5 [a] | 24.6 ± 0.1 [b] | 27.7 ± 0.1 [d] | 24.4 ± 0.5 [b] |
| | b* | 36.5 ± 0.2 [d] | 35.3 ± 0.3 [c] | 37.9 ± 0.3 [e] | 35.5 ± 0.3 [c] | 30.1 ± 0.6 [a] | 33.0 ± 0.2 [b] | 37.4 ± 0.1 [e] | 32.5 ± 0.5 [b] |

The placement of varying indexes such as a, b, c, d, e, f and g alongside the exact parameter values (as read for each row) indicates statistically significant differences in values at a level of $p < 0.05$. L* is color coefficient brightness, a* is color coefficient redness (+values)/greenness (-values); b* is color coefficient yellowness (+values)/blueness (-values).

The amount of soluble substances present in the juice is referred to as the extract value. During the fermentation process, there were slight variations in the extract values of the different juices. The mixture's extract values fell between those of beetroot juice and carrot juice, and this is because it is made up of 50% beetroot juice and 50% carrot juice. According to data from a table, beetroot juice contains more water-soluble substances than carrot juice [33,34]. When lactic acid bacteria are used during fermentation, reducing sugars and/or sucrose are converted to lactic acid, which is a water-soluble substance that can be detected through °Brix determination [35–37]. The gradual fermentation process ensures that there are no drastic changes in the extract content. However, the presence of lactic acid can replace the sugar content. The solid soluble content can vary depending on factors such as the type of cultivar, the harvest time, the maturity of the vegetable, and the prevailing weather conditions during its growth [38–41]. Studies conducted by Kazimierczak et al. [41] on dry matter in beetroot juices showed significantly lower values compared to our experiments. Additionally, the extract and density values presented in this study are higher than those in our previous study [29], which can be attributed to the maturity of the vegetables and the timing of the experiments.

Initial observations showed that carrot and mixed juice had the lowest density values. However, significant changes were observed after inoculation with LAB. *L. brevis* inoculated carrot juice recorded the lowest density value, while beetroot juice had the highest density value, regardless of the bacterial strain used (Tables 1 and 2). Minor changes in juice density were observed during the fermentation process, and no significant variation in density values was observed for any juice type or bacterial strain. Only carrot juice fermented with *L. brevis* showed significant changes in density during the final days of fermentation. However, there was no clear trend observed for this juice type or any other juice type analyzed. It is worth noting that density values are directly proportional to the juice's composition and dry matter content of the sample volume.

All samples showed an increase in total acidity, as seen in Figure 2. Juices fermented with *L. plantarum* had higher values compared to those fermented with *L. brevis*, indicating

faster lactic acid production by *L. plantarum* bacteria. It is important to note that this parameter measures the presence of titratable acids in the product and depends on the type and concentration of acids present in the raw material as well as those produced during fermentation [42,43]. The amount of lactic acid produced during fermentation depends on several factors, including the amount of available sugar, the type of lactic acid bacteria, and other substances present in the medium that may enhance or inhibit lactic acid production. The higher TA observed in Figure 2 for carrot juices may be due to the presence of malic acid, which can be converted to lactic acid during fermentation by some LAB strains [42,43]. Research involved heterofermentative strains such as *L. brevis* that produce acetic acid and ethanol as by-products, while facultatively heterofermentative strains such as *L. plantarum* use glucose to produce lactic acid [44].

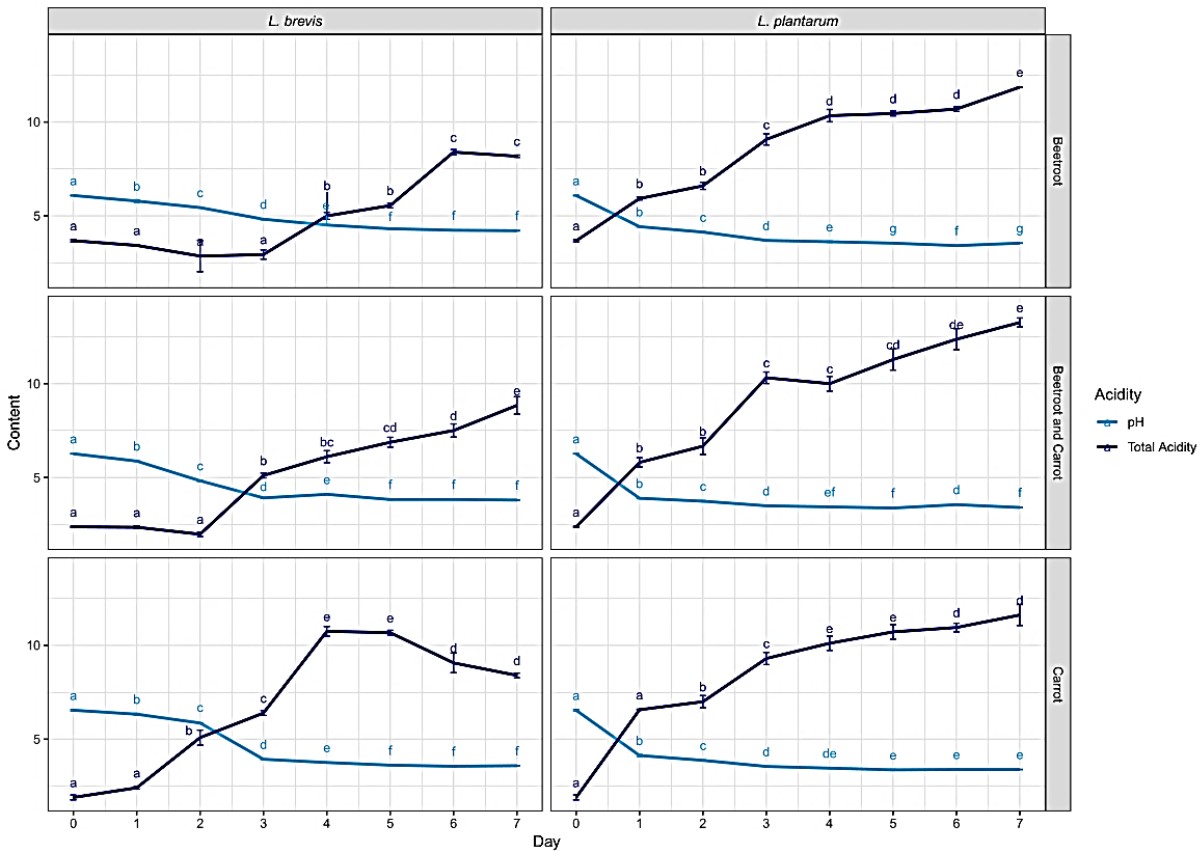

**Figure 2.** Total acidity (g lactic acid/100 g d.m.) and pH (-); Various indexes (such as a, b, c, d, e, f and g) within the series have displayed statistically significant differences in values at a level of $p < 0.05$.

During the fermentation process, all juices experienced a decrease in pH values on the following day, regardless of the juice type or bacterial strain used (Figure 2). However, it is crucial to note that juices inoculated with *L. plantarum* experienced a faster decrease in pH values compared to those inoculated with *L. brevis*. The lowest pH level of 3.3 was recorded for beetroot juice fermented with *L. plantarum* after 7 days, while carrot juices exhibited higher pH values than beetroot juices or their mixture. It is important to understand that the pH value directly corresponds to the lactic acid content produced during the growth of the LAB. Our research aligns with the findings of Marszałek et al. [45] for fresh beetroot juices and the findings of Tanguler et al. [46] for carrot juice, where similar pH values were observed. Moreover, it is worth noting that Tanguler et al. [46] observed the same trend with pH and total acidity as we observed in our research.

More obtained juices had good storage potential, while pH decreased below 4.5, which is a factor that determines the vegetable juices' stability [43].

In addition, a high correlation of total acidity with pH was observed, the faster the total acidity increased, the faster the pH decreased (Figure 2). This was confirmed by Pearson's correlation coefficient, which for that relation was for *L. plantarum* and *L. brevis*: for beetroot juices −0.96 and −0.65; for mixed juices −0.81 and −0.84; for carrot juice −0.94 and −0.92, respectively (Figure 3a–c).

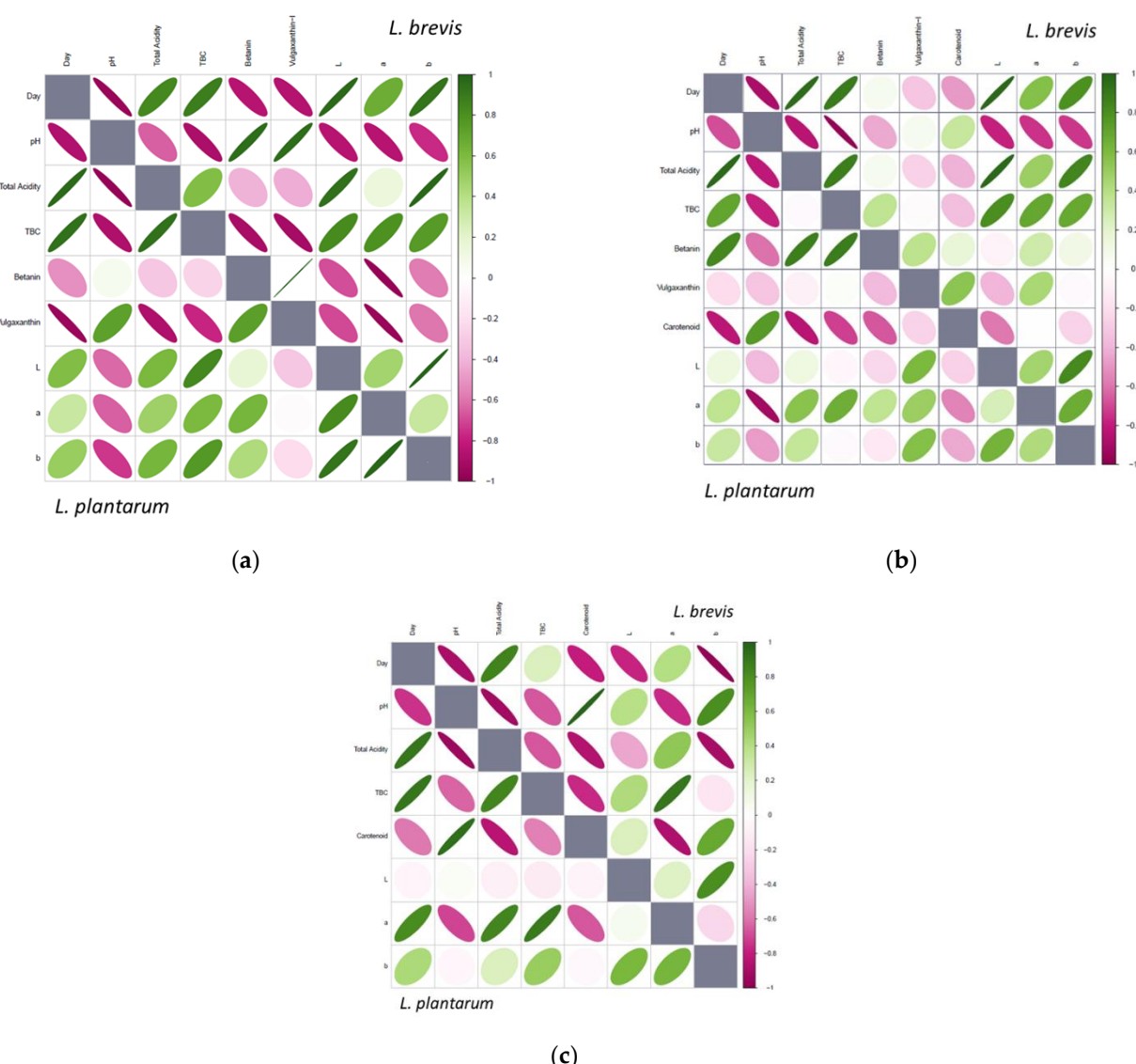

**Figure 3.** Correlations with selected parameters for (**a**) beetroot juices, (**b**) mixed juices and (**c**) carrot juices.

The color of a juice can be determined precisely by its L*, a* and b* values, which respectively represent brightness, red/green proportion, and yellow/blue proportion (refer to Tables 1 and 2). It is interesting to note that carrot juice has the highest brightness (with an L* value of 31–37), while beetroot juice is the darkest (with an L* value of 2.5–7.5). When beetroot juice is mixed with carrot juice in a 1:1 ratio, the resulting juice appears closer in brightness to beetroot juice than to carrot juice due to other color components that affect its overall color. Observations during the fermentation process of beetroot juice and its mixture with carrot juice showed changes in the redness and yellowness coefficients with no clear trend observed during the process. However, the post-fermentation values were higher than the baseline values in the juice. The type of strain used did not affect the

changes in color coefficients. The color coefficients are related to the pigment content and solid substances dissolved in juices [2,29,47,48].

A correlation was observed between color coefficients and the day of fermentation for juices inoculated with *L. brevis* (Pearson's correlation coefficient values higher than 0.6) (Figure 3a–c), which could also be related to the pigment behavior during the fermentation process (described in the next part of the article).

### 3.3. Pigment Behavior during Fermentation

Analysis of pigment content was made during each day of the fermentation process (Figure 4). Pigment identification was made for selected days (0, 3rd and 7th day of the process; Figure 5).

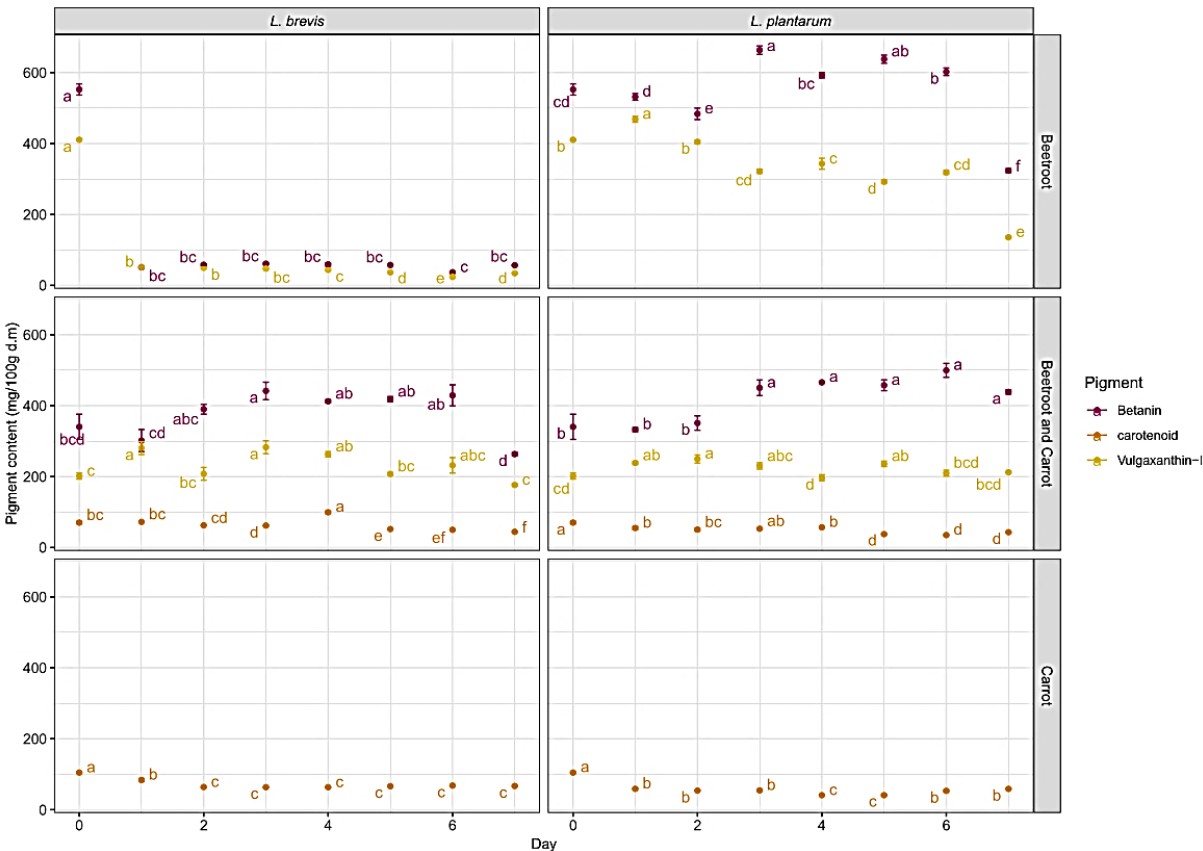

**Figure 4.** Kinetic behavior of pigment content. Various indexes (such as a, b, c, d, e and f) within the series have displayed statistically significant differences in values at a level of $p < 0.05$.

In the literature, pigments such as carotenoids and betalains are mentioned to be sensitive to low pH values. However, betalains show higher stability at low pH compared to carotenoids [4,37,49]. In the presented research, differentiation between lactic acid bacteria strains used on pigment behavior was observed. However, in all tested samples decreases in pigment values were observed.

During the fermentation process of beetroot juice with *L. brevis*, there was a significant reduction in the betalains content compared to the initial fermentation stage (Figure 4). Conversely, when using *L. plantarum*, the decrease in pigment components (red and yellow pigments) was slower and similar. The yellow pigments in beetroot juice exhibited a similar relationship to the red pigments. This could be due to the hydrolyzation process where glucose is obtained from betanin, which is a glucoside, and glucose and betanidin are created. These LAB strains can use this glucose and betanidin as a sugar source. Moreover, *L. brevis* exhibited a higher increase in bacterial count than *L. plantarum* in the juices, which could be attributed to the decrease in pigment rather than the total acidity and pH.

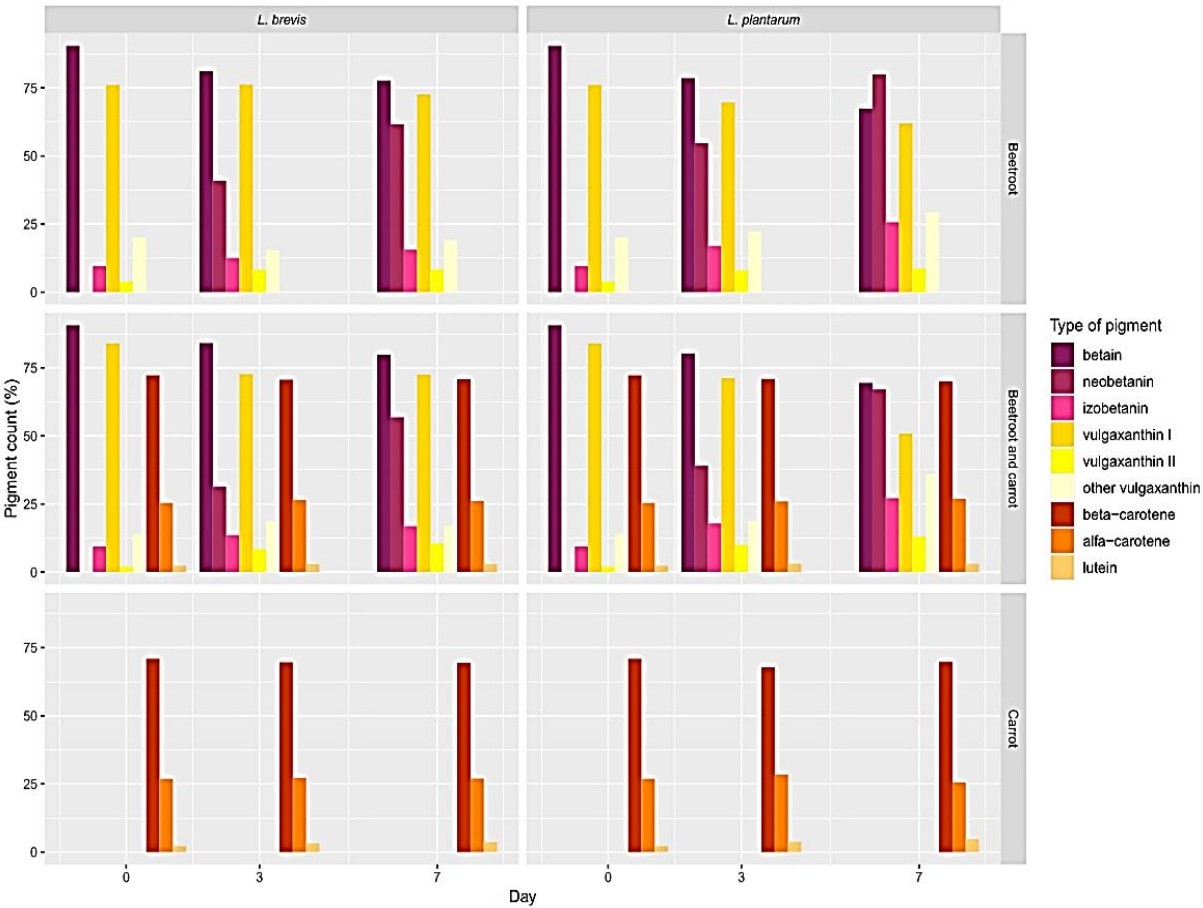

**Figure 5.** Pigment identification.

The presence of yellow coloration is associated with the presence of the head colorant vulgaxanthin. It was found in the tested products. Similar results were obtained for fermented beetroot juice by Czyżowska et al. [50] and Sawicki and Wiczkowski [51], while Czyżowska et al. [52] did not observe vulgaxanthin in their other study. In the study by Sawicki and Wiczkowski [51], this compound was gone after the 5th day of red beetroot juice fermentation.

The highest correlation was observed between betanin (red pigment), vulgaxanthin-I (yellow pigment), and the day of fermentation, which was related to the pH content (as shown in Figure 3a). Pearson's correlation coefficient values were found to be higher for *L. brevis* as compared to *L. plantarum*. For juices inoculated with *L. brevis*, the correlation between color coefficient and beetroot pigments was observed to be L* (−0.7), a* (−0.96), and b* (−0.57) (as depicted in Figure 3a).

The analysis conducted on the pigment compounds of beetroot juice prior to fermentation revealed the presence of betanin, isobetanin, vulgaxanthin-I, vulgaxanthin-II and other vulgaxanthin compounds (as depicted in Figure 5). Post-fermentation analysis indicated the presence of neobetanin. Across all samples, a slight decrease in betanin and vulgaxanthin-I was observed, while an increase in isobetanin and vulgaxanthin-II was noted. The fermentation process resulted in an increase in neobetanin, which is known to be formed from the double bond formation in betanin or isobetanin. This binding can occur at the same location and lead to the same product [53].

Significant decreases in carotenoid content were observed during the seven-day fermentation process of carrot juices. *L. plantarum* exhibited a higher decrease initially compared to *L. brevis*, but both strains displayed statistically significant decreases. Notably, juices with *L. brevis* remained stable after the second day of fermentation, while those with *L. plantarum* showed fluctuating values until the end of the process (see Figure 5).

Pigment compound analysis revealed the presence of β- and α-carotene, as well as a small amount of lutein (see Figure 5). Minor decreases were observed for β- and α-carotene during the process.

There were no significant correlations between the content of pigments and individual color factors, regardless of the tested juice. However, high values of correlation coefficients with pH values (positive) and total acidity values (negative) and an average correlation with the number of bacteria TBC (negative) were observed for carrot juices (Figure 3c).

The analysis of juices made from beetroot and carrot revealed that carotenoids behaved similarly regardless of the bacteria strain used. It is worth noting that betalain degradation was significantly lower in these juices. Interestingly, *L. plantarum* inoculated juices had higher values compared to those inoculated with *L. brevis*, as shown in Figure 4. All pigment compounds found in carrot and beetroot juices were also detected in mixed juices. Moreover, after fermentation, neobetanin was detected in all fermented mixed juices, which was absent in fresh juice, as depicted in Figure 5. This strongly suggests that the high antioxidant properties of carotenoid pigments may be the key factor responsible for the improved stability of betalain pigments in mixed juices.

In mixed juices, as in the case of carrot juices, no correlation was observed between color coefficients and individual types of pigments. Only in the case of the analysis of the influence of the fermentation process on the content of pigments, carotenoids (negative) and betanin (negative) shown, however, only when *L. plantarum* was used as a strain, for *L. brevis* such relationships were not observed (Figure 3b).

## 4. Conclusions

Based on the results of our analysis, it can be concluded that *Lactiplantibacillus plantarum* and *Levilactobacillus brevis* LAB strains are capable of fermenting beetroot, carrot, and a combination of these juices. *Lactiplantibacillus plantarum* was demonstrated to be the superior strain for vegetable juice fermentation, with higher measurement stability and faster fermentation time (pH values reached the desired level more quickly). Additionally, vegetable juices fermented with this strain retained high and stable levels of betalain pigments. On the other hand, *Levilactobacillus brevis* proved to be an inferior strain for vegetable juice fermentation, with measurements that were often unstable and slower lactic fermentation. However, this strain had a lower impact on carotenoids, which are pigments insoluble in water, than *Lactiplantibacillus plantarum*. Conversely, betalain, a water-soluble pigment with nitrogen in its structure, was more influenced by *Lactiplantibacillus plantarum*. The most stable juice was the mixture, as it had a higher bacteria count and pigment content. Future tests could explore other LAB strains and shorten the fermentation time to five days.

**Author Contributions:** Conceptualization, E.J.-T. and K.P.; methodology, E.J.-T., K.P., K.R. and Ł.W.; validation, E.J.-T., K.P., K.R. and Ł.W.; formal analysis, E.J.-T., K.P., Ł.W. and U.T.; investigation E.J.-T., K.P. and K.R.; resources, E.J.-T. and K.P.; data curation, E.J.-T., K.P., K.R., A.S. and Ł.W.; writing—original draft preparation, E.J.-T., K.P., K.R., A.S., Ł.W., U.T., D.W.-R. and M.G.; writing—after revision E.J.-T., K.P., K.R., A.S., Ł.W., U.T., D.W.-R. and M.G.; visualization, E.J.-T., K.P. and A.S.; supervision, E.J.-T. and D.W.-R.; All authors have read and agreed to the published version of the manuscript.

**Funding:** This research received no external funding.

**Institutional Review Board Statement:** Not applicable.

**Informed Consent Statement:** Not applicable.

**Data Availability Statement:** Data available from the corresponding author.

**Acknowledgments:** The authors would like to thank Tomasz Zakrzewski and Adriana Kuć for their help in carrying out some of the analyses. Research equipment was purchased as part of the "Food and Nutrition Centre—modernisation of the WULS campus to create a Food and Nutrition Research and Development Centre (CŻiŻ)" co-financed by the European Union from the European Regional

Development Fund under the Regional Operational Programme of the Mazowieckie Voivodeship for 2014–2020 (Project No. RPMA.01.01.00-14-8276/17).

**Conflicts of Interest:** The authors declare no conflict of interest.

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
