# Peer review of "Changes in Physical and Chemical Parameters of Beetroot and Carrot Juices Obtained by Lactic Fermentation"

_applsci, doi:10.3390/app13106113_

Round 1

Reviewer 1 Report

The manuscript seems fine but I need to make some recommendations

First, the introduction section has been well written but, some of the points had been passed.

I strongly recommend that the authors should cite the references shown below which were directly related to a fermented vegetable juice called shalgam. 

References:

A.E. YETİMAN, F. ORTAKCI, Genomic, probiotic, and metabolic potentials of Liquorilactobacillus nagelii AGA58, a novel bacteriocinogenic motile strain isolated from lactic acid-fermented shalgam, JOURNAL OF BIOSCIENCE AND BIOENGINEERING, 2023, 135(1), 34-43.

 A.E. YETİMAN, M. HORZUM, D. BAHAR, M. AKBULUT,  Assessment of Genomic and Metabolic Characteristics of Cholesterol-Reducing and GABA Producer Limosilactobacillus fermentum AGA52 Isolated from Lactic Acid Fermented Shalgam Based on “In Silico” and “In Vitro” Approaches. PROBİOTİCS AND ANTİMİCROBIAL PROTEINS, 2023. https://doi.org/10.1007/s12602-022-10038-2.

A.E. YETIMAN, A. KESKIN, B. N. DARENDELI, S. E. KOTIL, F. ORTAKCI & M. DOGAN, Characterization of genomic, physiological, and probiotic features Lactiplantibacillus plantarum DY46 strain isolated from traditional lactic acid  fermented shalgam beverage, FOOD BIOSCIENCE, 2022, 46, 101499.

In Table 1, the authors should use upperscript for letter ranking

table 1 has been published twice, please fix it

In figure3, 5 and 6 error bars are missing, please fix it

Please remove image 2 as it does not enhance the ingredients of the manuscript. Neither the API test for carbohydrate fermentation patterns nor their information regarding the 16s rDNA sequence confirmation of the strains were employed. It was going to be more appropriate than the one we currently have instead of this. 

Author Response

Comments and Suggestions for Authors

Dear Reviewer 1

We would like to thank you for all your comments on the manuscript, all of them have improved our work. We hope that answers and corrections will satisfy you.

With best regards, on behalf of all authors

Dorota Witrowa-Rajchert

Reviewer 1

The manuscript seems fine but I need to make some recommendations First, the introduction section has been well written but, some of the points had been passed

I strongly recommend that the authors should cite the references shown below which were directly related to a fermented vegetable juice called shalgam. 

References:

A.E. YETİMAN, F. ORTAKCI, Genomic, probiotic, and metabolic potentials of Liquorilactobacillus nagelii AGA58, a novel bacteriocinogenic motile strain isolated from lactic acid-fermented shalgam, JOURNAL OF BIOSCIENCE AND BIOENGINEERING, 2023, 135(1), 34-43.

 A.E. YETİMAN, M. HORZUM, D. BAHAR, M. AKBULUT,  Assessment of Genomic and Metabolic Characteristics of Cholesterol-Reducing and GABA Producer Limosilactobacillus fermentum AGA52 Isolated from Lactic Acid Fermented Shalgam Based on “In Silico” and “In Vitro” Approaches. PROBİOTİCS AND ANTİMİCROBIAL PROTEINS, 2023. https://doi.org/10.1007/s12602-022-10038-2.

A.E. YETIMAN, A. KESKIN, B. N. DARENDELI, S. E. KOTIL, F. ORTAKCI & M. DOGAN, Characterization of genomic, physiological, and probiotic features Lactiplantibacillus plantarum DY46 strain isolated from traditional lactic acid  fermented shalgam beverage, FOOD BIOSCIENCE, 2022, 46, 101499.

Answer: Thank you for that suggestion. We have read those articles and cited in our manuscript.

In Table 1, the authors should use upperscript for letter ranking

Answer: It has been corrected.

table 1 has been published twice, please fix it

Answer: It should be two tables, it has been named correctly as Table 1 and Table 2. The first one is for L. plantarum while the second one is for L. brevis

In figure3, 5 and 6 error bars are missing, please fix it

Answer: In the pictures error bars are placed, but they are so small that it is hard to see them.

Please remove image 2 as it does not enhance the ingredients of the manuscript. Neither the API test for carbohydrate fermentation patterns nor their information regarding the 16s rDNA sequence confirmation of the strains were employed. It was going to be more appropriate than the one we currently have instead of this

Answer: Image 2 has been removed. We thought placing it in the manuscript could visualize the difference between those two used bacteria.

We did not do bacterial identification because the strains used in the study were from a collection of pure cultures.

If we would do a study involving autochthonous microflora we will try to do the suggested tests. Thank you for that sugestion.

Reviewer 2 Report

Recommendation: Minor

The manuscript Changes in physical and chemical parameters of beetroot and carrot juices obtained by lactic fermentation, the methodology was reasonable and technically sound.

Comments to the Author:

The main procedure and findings of the study are well expressed. Introduction: A brief survey of existing literature, the purpose, importance, and innovation of the research is well mentioned.

Below are some important suggestions.

Point 1. add numerical results in the abstract

Point 2 Make suggestions for future study in the abstract

Point 3. In the introduction part of the article, a lot of information is given, if possible, the literature should be briefly explained.

Point 4. The norm of temperature and time applied to juices is very high, is this true?

Point 5. Make suggestions for future studies in the conclusion sentence.

Author Response

Comments and Suggestions for Authors

Dear Reviewer 2

We would like to thank you for all your comments on the manuscript, all of them have improved our work. We hope that answers and corrections will satisfy you.

With best regards, on behalf of all authors

Dorota Witrowa-Rajchert

Recommendation: Minor

The manuscript Changes in physical and chemical parameters of beetroot and carrot juices obtained by lactic fermentation, the methodology was reasonable and technically sound.

Comments to the Author:

The main procedure and findings of the study are well expressed. Introduction: A brief survey of existing literature, the purpose, importance, and innovation of the research is well mentioned.

Point 1. Add numerical results in the abstract.

It has been added.

Point 2 Make suggestions for future study in the abstract.

It has been added.

Point 3. In the introduction part of the article, a lot of information is given, if possible, the literature should be briefly explained.

We have shortened the introduction and the overall information has been reorganized to summarize it.

Point 4. The norm of temperature and time applied to juices is very high, is this true?

We have in article two temperatures range:

1.      temperature of 80°C was used in the pasteurization process. In this process, we need to degrade indigenous bacteria, as well as spores, and mold spores. As a result of previous studies, this temperature (80°C) was sufficient for this degradation, at the same time, in the juice we did not observe physical changes and degradation of the dye. Lower temperatures did not guarantee the degradation of spores, which could result in misinterpretation of results with dedicated bacterial strains.

2.      If the question is about the fermentation temperature of 28°C, this was the optimal temperature for these bacteria for growth and fermentation.

Point 5. Make suggestions for future studies in the conclusion sentence.

It has been added.

Reviewer 3 Report

The authors studied the lactic acid fermentation of beetroot and carrot juices and observed the changes in some parameters. Recently, it is a very interesting area in nutrient supplements for all-aged people.

However, the manuscript is unfortunately very confusing and boring.

Please describe clearly the small letters (a, b, c, d) of Figures 1 and 2 as a footnote.

Please describe clearly the L*, a*, and b* of Tables 1 and 2 as a footnote.

Line 240, 241, 321, 322

Are these sentences the figure captions or not?

Please describe what are these.

Line 253~259

Why did the authors write the sentences in italic?

Please describe the reasons.

Author Response

Comments and Suggestions for Authors

Dear Reviewer 3

We would like to thank you for all your comments on the manuscript, all of them have improved our work. We hope that answers and corrections will satisfy you.

Thank you for your time and consideration.

Best regards, on behalf of all authors

Dorota Witrowa-Rajchert

Comments and Suggestions for Authors

The authors studied the lactic acid fermentation of beetroot and carrot juices and observed the changes in some parameters. Recently, it is a very interesting area in nutrient supplements for all-aged people.

However, the manuscript is unfortunately very confusing and boring

Dear reviewer,

We have revised the work by rewriting the sentences to make them more readable. Our aim was to improve the clarity and organization of the news to avoid confusion and chaos in the descriptions. We hope that the corrections we made will allow for greater readability and understanding of the article.

Please describe clearly the small letters (a, b, c, d) of Figures 1 and 2 as a footnote.

Please describe clearly the L*, a*, and b* of Tables 1 and 2 as a footnote.

I have made the necessary additions. The information regarding letters has been modified, and a description of color coefficients L*, a*, and b* has been included below Tables 1 and 2.

Line 240, 241, 321, 322, Are these sentences the figure captions or not? Please describe what are these.

The provided sentences pertain to the statistical treatment details that are depicted in the accompanying pictures. We have integrated these sentences with the figure caption in the preceding line.

Line 253~259, Why did the authors write the sentences in italic?  Please describe the reasons.

I apologize for the mistake. Thank you for bringing it to my attention. The issue has been resolved, and only the names of bacteria are now in italics.

Round 2

Reviewer 3 Report

Where is the figure caption?

Please see an attached file.

Author Response

Dear reviewer,

We have taken action on the recommendation of another reviewer and removed a figure, along with its caption. As proof of our action, we have left a crossed-out image. Thank you for your assistance in ensuring the quality of our work.

Best regards,

Dorota Witrowa-Rajchert